# Transglutaminase 2-Mediated p53 Depletion Promotes Angiogenesis by Increasing HIF-1α-p300 Binding in Renal Cell Carcinoma

**DOI:** 10.3390/ijms21145042

**Published:** 2020-07-17

**Authors:** Seon-Hyeong Lee, Joon Hee Kang, Ji Sun Ha, Jae-Seon Lee, Su-Jin Oh, Hyun-Jung Choi, Jaewhan Song, Soo-Youl Kim

**Affiliations:** 1Division of Cancer Biology, Research Institute, National Cancer Center, Goyang, Gyeonggi-do 10408, Korea; shlee1987@gmail.com (S.-H.L.); wnsl2820@gmail.com (J.H.K.); jsha9595@gmail.com (J.S.H.); ljs891109@gmail.com (J.-S.L.); sujiniii225@gmail.com (S.-J.O.); labchoihj@gmail.com (H.-J.C.); 2Department of Biochemistry, College of Life Science and Biotechnology, Yonsei University, Seoul 03722, Korea; jso678@yonsei.ac.kr

**Keywords:** transglutaminase 2, angiogenesis, HIF-1α, p53, renal cell carcinoma

## Abstract

Angiogenesis and the expression of vascular endothelial growth factor (VEGF) are increased in renal cell carcinoma (RCC). Transglutaminase 2 (TGase 2), which promotes angiogenesis in endothelial cells during wound healing, is upregulated in RCC. Tumor angiogenesis involves three domains: cancer cells, the extracellular matrix, and endothelial cells. TGase 2 stabilizes VEGF in the extracellular matrix and promotes VEGFR-2 nuclear translocation in endothelial cells. However, the role of TGase 2 in angiogenesis in the cancer cell domain remains unclear. Hypoxia-inducible factor (HIF)-1α-mediated VEGF production underlies the induction of angiogenesis in cancer cells. In this study, we show that p53 downregulated HIF-1α in RCC, and p53 overexpression decreased VEGF production. Increased TGase 2 promoted angiogenesis by inducing p53 degradation, leading to the activation of HIF-1α. The interaction of HIF-1α and p53 with the cofactor p300 is required for stable transcriptional activation. We found that TGase 2-mediated p53 depletion increased the availability of p300 for HIF-1α-p300 binding. A preclinical xenograft model suggested that TGase 2 inhibition can reverse angiogenesis in RCC.

## 1. Introduction

The pathologic phenotype of renal cell carcinoma (RCC) is characterized by abundant neovascularization and arteriovenous fistula formation [1]. Vascular endothelial growth factor (VEGF) is critical for angiogenesis; tumor-induced angiogenesis is mediated by the paracrine action of VEGF produced by tumor cells, and VEGFR is expressed in vascular endothelial cells [2]. VEGF mRNA expression is 3–13-fold higher [3], and VEGF protein expression is 3–37-fold higher in RCC tumors than in the normal parenchyma [4]. These data are consistent with the RCC phenotype of high neovascularization. Analysis of mRNA and protein expression in RCC patients from The Cancer Genome Atlas shows that transglutaminase 2 (TGase 2) is upregulated in RCC [5,6], and TGase 2 upregulation is associated with increased metastatic potential and worse prognosis [5]. TGase 2 catalyzes the formation of Nε-(γ-glutamyl)-lysine cross-links (E.C. 2.3.2.13) between the γ-carboxamide groups of peptide-bound glutamine residues and the γ-amino groups of protein- and peptide-bound lysine residues [7]. TGase 2 plays an important role in the regulation of p53 in RCC (reviewed in [8]).

A meta-analysis of RCC patients showed that increased levels of TGase 2 are inversely correlated with 5-year disease-free survival [9]. TGase 2 is upregulated in cells migrating to fibrin clots and contributes to extracellular matrix (ECM) remodeling, as well as promoting neovascularization, supporting a role for TGase 2 in wound healing and angiogenesis [10]. Three cellular domains are involved in tumor angiogenesis: endothelial cells, the ECM, and cancer cells. In endothelial cells, TGase 2 plays a role in the cross-linking between VEGFR-2 and the α_v_β_3_ integrin on the surface of endothelial cells; this stabilizes VEGFR-2 signaling and promotes the nuclear translocation of VEGFR-2 by catalyzing the formation of the TGase 2-VEGFR2 complex, thereby promoting endothelial cell migration [11]. In the ECM, the proposed mechanism is that extracellular TGase 2 binds directly to the angiogenesis inhibitor endostatin to neutralize its inhibitory effect [12]. Endostatin inhibits the binding of VEGF_165_ to VEGFR-2 on endothelial cells [12]. Heparin or the heparan sulfate-binding domain of endostatin at Arg^27^ and Arg^139^ binds to the C-terminal TGase 2 GTP-binding site at Arg^580^ and Arg^476^~Ser^483^ [13]. The binding of endostatin to TGase 2 inhibits the interaction between endostatin and heparan sulfate; however, TGase 2 maintains the ability to bind to fibronectin in the ECM through the gelatin-binding domain at residues 550–565 [14], through the N-terminal TGase 2 fibronectin-binding domain at Trp^88^–Tyr^106^ [15], or at Lys^30^, His^116^, or Arg^134^ [16]. Therefore, extracellular TGase 2 promotes VEGF signaling by decreasing endostatin activity. In cancer cells, the role of TGase 2 remains unclear, and there are no studies investigating how TGase 2 is involved in the regulation of VEGF, which mediates angiogenesis in RCC. In this study, we examined the role of TGase 2 in angiogenesis and explored the therapeutic potential of TGase 2 inhibition as an anti-angiogenic strategy in RCC.

## 2. Results

### 2.1. TGase 2 Expression Is Correlated with the Expression Level of CD31

Immunohistochemical analysis of TGase 2 and CD31 was performed using a tissue microarray (41 cases). The analysis detected cases of low TGase 2 expression, high TGase 2 expression only in the cytoplasm, high TGase 2 expression only in the cytoplasmic membrane, and high TGase 2 expression in both the cytoplasm and the cytoplasmic membrane (Figure 1A and Appendix A). Six normal kidney tissue samples from the glomerulus, proximal tubule, and blood vessels showed low TGase 2 expression (Figure 1B). TGase 2 was expressed at high levels in 24 of the 41 RCC cases (58.5%), of which 17 showed TGase 2 expression only in the cytoplasmic membrane (41.5%). In three RCC cases, TGase 2 was highly expressed only in the cytoplasm (7.3%). In four RCC cases, TGase 2 was expressed in both the cytoplasm and the cytoplasmic membrane (9.7%) (Figure 1A,C). 

To examine the association of TGase 2 with angiogenesis, TGase 2 expression was correlated with the number of cells positive for the endothelial cell marker CD31. Cases in which TGase 2 expression was restricted to the cytoplasmic membrane showed an approximately 4.3-fold higher number of CD31-positive cells than normal kidney tissues and RCC with low TGase 2 expression (Figure 1C). As significant correlation was identified in the expression levels of TGase 2 and CD31 in kidney renal clear cell carcinoma, the GEPIA database was used in the present study to analyze the correlations between *TGM2* and *PECAM1*. TGase 2 is encoded by the *TGM2* gene and CD31 is encoded by the *PECAM1* gene in humans. The results revealed that expression levels of *TGM2* and *PECAM1* were positively correlated (*p*-value = 3.5e-12, *R* = 0.3) (Figure 1D).

### 2.2. TGase 2 Inhibition Induces p53-Dependent Downregulation of Hypoxia-Inducible Factor (HIF)-1α

Tumor-suppressor genes such as p53 and von-Hippel Lindau (VHL) regulate the levels and activity of HIF-1α. p53 inhibits HIF-1 activity by targeting the HIF-1α subunit for mouse double minute 2 homolog (MDM2)-mediated ubiquitination and proteasomal degradation [17]. The TGase 2 inhibitor streptonigrin stabilizes p53-mediated apoptosis and inhibits tumor growth in vivo [6]. Since p53 downregulates HIF-1α expression, we hypothesized that streptonigrin may decrease the levels of HIF-1α. The results showed that hypoxia upregulated HIF-1α protein expression in CAKI-1 and ACHN cells, and streptonigrin upregulated p53 under hypoxic conditions. Streptonigrin significantly downregulated HIF-1α in a dose-dependent manner, whereas it had no effect on negative regulators of HIF-1α such as VHL and MDM2 (Figure 2A,B). Similar results showing that streptonigrin downregulated HIF-1α protein expression were obtained in cells exposed to cobalt chloride (CoCl_2_)-induced chemical hypoxia (Figure 2C,D). The effect of streptonigrin on the translocation of p53 and HIF-1α was examined under conditions of hypoxia. The results showed that hypoxia increased the nuclear levels of p53 and HIF-1α, whereas streptonigrin treatment further increased nuclear p53 and downregulated nuclear HIF-1α under hypoxic conditions (Figure 2E,F). Taken together, these results indicate that TGase 2 inhibition is involved in the regulation of HIF-1α under hypoxia. 

### 2.3. Competition between p53 and HIF-1α for Binding to p300

The transcriptional activity of HIF-1α and p53 requires their interaction with the co-activator p300, and the transactivation or transcriptional repression between the two factors depends on the availability of p300 [18,19]. To test whether the TGase 2 inhibition-induced increase of apoptosis [6] was attributed to increased p53–p300 binding, ACHN extracts treated with streptonigrin for 4 h under hypoxia were immunoprecipitated for p300 and immunoblotted for p53 (Figure 3A,B). The treatment of ACHN cells with streptonigrin at 0.5 mM caused a 2.5-fold increase in p300 binding to p53 (Figure 3A,B), indicating that TGase 2 inhibition promoted the p300–p53 interaction and decreased the p300–HIF-1α interaction. To test the effect of MDM2 and TGase 2 expression on the increase in p53 binding to p300, TGase 2 and MDM2 mRNA expression was measured by RT-PCR in cells overexpressing p53 and/or exposed to hypoxia for 4 h (Figure 3C). The results showed that p53 transfection or hypoxia had no effect on the transcription of MDM2 or TGase 2 (Figure 3C).

### 2.4. TGase 2 and p53 Modulate VEGF Secretion under Hypoxia

To examine the relation between p53, TGase 2, and VEGF regulation under hypoxia, VEGF was detected by ELISA in cell supernatants. The level of secreted VEGF was higher under hypoxic than under normoxic conditions. Streptonigrin treatment significantly decreased VEGF secretion in cells under hypoxia (Figure 4A). This was confirmed by the siRNA-mediated silencing of TGase 2, which significantly decreased VEGF secretion in cells under hypoxia (Figure 4B). The effect of p53 on VEGF was confirmed by p53 wild-type transfection. p53 overexpression significantly decreased VEGF secretion in cells under hypoxia (Figure 4C). 

### 2.5. Pazopanib Combined with Streptonigrin Reduces Tumor Vascular Density

To test the activity of anti-angiogenic drug such as pazopanib combined with streptonigrin in RCC in vivo, BALB/c-nude mice were injected subcutaneously with CAKI-1 cells. Pazopanib is a potent and selective multi-targeted receptor tyrosine kinase inhibitor that blocks tumor growth and inhibits angiogenesis. Mice were randomized to receive vehicle alone, pazopanib (100 mg/kg), streptonigrin (0.1 mg/kg), or a combination of both drugs every 2 days by oral gavage. After 43 days, pazopanib, streptonigrin, and the combination of both drugs decreased the mean tumor volume by 17.7%, 62.5%, and 65.5% (Figure 5A) and the mean tumor weight by 12.5%, 58.8%, and 58.8% (Figure 5B), respectively, compared with the vehicle-treated group. No signs of toxicity or changes in mouse body weight were observed (Figure 5C). To evaluate tumor angiogenesis, tissues were subjected to immunohistochemical staining for the endothelial marker CD31. Pazopanib, streptonigrin, and the combination of both drugs decreased the mean number of CD31-positive cells by 26.5%, 30.1%, and 50.4%, respectively, compared with the control group, indicating that the combination treatment had the strongest effect on inhibiting tumor angiogenesis (Figure 5D,E and Appendix A). Although the combination treatment was the most effective for reducing tumor angiogenesis, tumor growth by a combination treatment of pazopanib and streptonigrin was not reduced synergistically (Figure 5A,E). This suggests that pazopanib decreased tumor angiogenesis without affecting tumor growth, whereas streptonigrin decreased both tumor angiogenesis and tumor growth through p53 activation [6].

## 3. Discussion

In this study, we showed that TGase 2 promotes proangiogenic response in RCC by activating HIF-1α through the regulation of p53. TGase 2 binding to p53 activated HIF-1α by promoting the HIF-1α–p300 interaction. Streptonigrin inactivated HIF-1α by promoting the release of p53 from binding to TGase 2, thereby increasing p53–p300 binding, which decreased the amount of secreted VEGF. The VEGF inhibitor pazopanib did not have a synergistic effect with streptonigrin (Figure 6). TGase 2 inhibition had a dominant inhibitory effect on anti-angiogenesis mediated by the regulation of HIF-1α in cancer cells. 

p53 deletion promotes tumor growth and neovascularization in a xenograft model [20]. p53 downregulates HIF-1α under hypoxia by inducing the MDM2-mediated proteasomal degradation of HIF-1α in colon cancer cells [20]. Another proposed mechanism is that the limited amount of CREB-binding protein (CBP or p300) inactivates HIF-1α because p53 binds competitively to the CH1 domain of the p300 co-activator [19,21,22], and p53 activity is synergistically activated by p300 binding [22,23] (Figure 6). p53 with mutations at the N-terminal transactivating domain (22/23 residue) cannot inhibit HIF-1α activity in breast cancer cells because the mutations prevent p53 binding to p300 [21,24]. Therefore, low levels of p53 inactivate HIF-1α transactivation by competing for the p300 cofactor, whereas high levels of p53 promote MDM2-mediated ubiquitination and the degradation of HIF-1α [19] (Figure 6). The tumor suppressor *p53* is one of the most mutated genes in human cancer [25]. However, the p53 mutation rate in RCC is only 4% [26]. Stabilized p53 in RCC can inactivate HIF-1α by inducing MDM2-mediated HIF-1α degradation or through p300 competitive binding-mediated HIF-1α inhibition (Figure 6). The N-terminal transactivation domain of p53 binds efficiently to the N-terminal region of TGase 2 [26]. Increased levels of TGase 2 deplete p53 by promoting autophagic degradation through the formation of a triple complex of p53-TGase 2-p62 [26], which may activate HIF-1α for angiogenesis by promoting HIF-1α-p300 binding. Streptonigrin interferes with p53 binding to TGase 2 through competitive binding to TGase 2 in RCC [6]. Therefore, streptonigrin treatment may promote p53–p300 binding by interfering with p53–TGase 2 binding, which inactivates HIF-1α by decreasing the level of free p300 (Figure 6). In RCC cell lines, siRNA-mediated TGase 2 knockdown induces cell death, whereas TGase 2 knockdown does not induce cell death in the normal immortalized HEK293 cell line [6,26,27,28,29]. This implies that RCC specifically relies on TGase 2 for survival. Several pathways may underlie the constitutive upregulation of TGase 2 expression in RCC [8]. TGase 2 acts as a chaperone protein in addition to its role as a cross-linking enzyme [30]. TGase 2 forms a ternary complex wherein the N-terminus of p53 binds to the N-terminus of TGase 2, while the C-terminus of TGase 2 binds to the N-terminus of p63 [26]. Therefore, TGase 2 is a major regulator of p53 in RCC. We demonstrated that blocking TGase 2 binding to p53 with streptonigrin stabilized p53, which resulted in apoptosis induction [6] (Figure 6).

In this study, we elucidated two important roles of TGase 2. First, TGase 2 inhibition induces apoptosis by promoting p53–p300 binding. Second, increased p53–p300 binding inactivates HIF-1α, thereby suppressing angiogenesis. 

TGase 2 increases angiogenesis in the ECM domain by binding directly to endostatin or VEGF. Endostatin is a C-terminal peptide of collagen XVIII that inhibits angiogenesis and tumor growth [31]. Analysis of the endostatin interaction network identified several proteins, including glycosaminoglycans, matricellular proteins, collagens, and TGase 2 [12]. Endostatin cannot interact simultaneously with TGase 2 and heparan sulfate, because they share the binding site on endostatin [13]. Endostatin binding to TGase 2 is very strong, showing an affinity in the nanomolar range [13], whereas endostatin binding to heparan sulfate glycosaminoglycans is weak [32]. This suggests that TGase 2 blocks the endostatin-mediated inhibition of VEGF–heparan sulfate–VEGFR interaction, which promotes VEGF signaling. 

The role of TGase 2 in the ECM domain under non-conventional VEGF signaling was recently proposed [33]. TGase 2 binds to VEGF to form a 90 kDa VEGF–TGase 2 complex in the ECM and interacts with HSP90 on the surface of microvesicles secreted from cancer cells, which promotes VEGF–VEGFR interaction [33]. This study showed that the 90 kDa form of VEGF is resistant to bevacizumab because binding of the complex to HSP90 on the surface of microvesicles protects it from bevacizumab. However, treatment with an HSP90 inhibitor restored sensitivity to bevacizumab by inhibiting the binding of the TGase 2–VEGF complex to HSP90 [33].

The effect of TGase 2 on placental damage in celiac disease provides further evidence of its involvement in angiogenesis [34]. Celiac disease, an autoimmune enteropathy caused by a gluten diet, is characterized by the presence of autoantibodies against TGase 2 [35]. Autoantibodies against TGase 2 bind to human endometrial endothelial cells and suppress new vessel formation [35]. This may be associated with reduced fertility and an increased risk of adverse pregnancy events such as miscarriages in patients with celiac disease [36]. The autoantibody epitope against TGase 2 was identified as the N-terminus of TGase 2 at Arg^116^ and His^134^ [37]. Autoantibodies block the matrix-binding domain of TGase 2, decreasing co-localization with VEGFR-2 and the activation of VEGFR-2 signaling, which inhibits angiogenesis. 

## 4. Materials and Methods 

### 4.1. Antibodies and Reagents

Anti-TGase 2 antibody (Cat. No. MA5-12739; 1:1000) was purchased from Sigma-Aldrich (St. Louis, MO, USA). Antibodies against p53 (Cat. No. sc-126; 1:500), MDM2 (Cat. No. sc-813; 1:500), p300 (Cat. No. sc-584; 1:500), Lamin B (Cat. No. sc-6216; 1:1000), GAPDH (Cat. No. sc-47724; A:1000), and β-actin (Cat. No. sc-47778; 1:500) were purchased from Santa Cruz Biotechnology (Dallas, TX, USA). Anti-HIF-1α (Cat. No. A700-001; 1:1,000) was purchased from Bethyl laboratories (Montgomery, TX, USA). Anti-VHL (Cat. No. MS-690-P0) was purchased from Thermo Scientific (Waltham, MA, USA). Anti-CD31 (Cat. No. ab28364) was purchased from Abcam (Cambridge, UK). Streptonigrin (Cat. No. S1014) and cobalt chloride (Cat. No. C8661) were purchased from Sigma-Aldrich. Pazopanib hydrochloride salt (Cat. No. P6755) was purchased from LC Laboratories (Woburn, MA, USA). INTERFERin (Cat. No. 409-50) and jetPEI (Cat. No. 101-40N) transfection reagents were from Polyplus (NY, USA). A small interfering RNA (siRNA) duplex targeting human TGase 2 was obtained from GenePharma (Shanghai, China). TGM2 siRNA, 5’-UGG GCA GUU UGA AGA UGG GAU CCU A-3’; and control siRNA, 5’-CCU CGU GCC GUU CCA UCA GGU AGU U-3’. p3XFLAG p53 plasmid was described previously [26].

### 4.2. Cell Culture

CAKI-1 and ACHN cells were purchased from ATCC (Manassas, VA, USA). Cells were cultured in complete DMEM (Hyclone, Marlborough, MA, USA) containing 10% FBS (Hyclone) and penicillin/streptomycin in an atmosphere of 5% CO_2_/100% humidity at 37°C. Cells were incubated for 4 h under normoxic condition (5% CO_2_) using conventional incubator or hypoxic conditions (1% O_2_, 5% CO_2_, and 94% N_2_) using a Whitley H35 Hypoxystation (Don Whitley Scientific, West Yorkshire, UK). Cells were incubated with or without 500 μM CoCl_2_ for 24 h.

### 4.3. Western Blotting

For Western blot analysis, whole cell lysates were prepared using RIPA buffer (Cat. No. R0278, Sigma-Aldrich) supplemented with a protease inhibitor cocktail. Nuclear and cytoplasmic extraction was performed using the NE-PER reagent kit (Cat. No. 78835; Thermo Scientific) supplemented with a protease inhibitor cocktail. Protein was quantified with the BCA protein assay (Cat. No. 23225; Thermo Scientific) to normalize protein expression. For immunoprecipitation, cell lysates were mixed with antibodies and Protein A/G UltraLink Resin (Cat. No. 53132; Thermo Scientific) and incubated at 4 °C overnight. Immunoprecipitates were washed with lysis buffer before Western blot analysis. Proteins were resolved by SDS-PAGE and transferred to PVDF membranes. The membranes were incubated with 5% BSA plus 0.1% Tween 20 for 1 h at room temperature and then incubated with primary antibodies at 4 °C overnight. The membranes were washed with TBST for 1 h and incubated with HRP-conjugated secondary antibodies for 1 h at room temperature. After washing the membranes with TBST for 1 h, chemiluminescence detection was performed using a Western blotting substrate (Cat. No. LF-QC0101; Seoul, Korea), followed by imaging with FUSION-Solo (VILBER Lourmat, FRA). Western blot band quantification was performed using ImageJ. 

### 4.4. Real-Time PCR

Total RNA was extracted from the different cell lines using the TRIzol^®^ Reagent (Life Technologies, Carlsbad, CA, USA) according to the manufacturer’s protocols. RNA concentration was determined by measuring the absorbance at 260 nm using a Nanodrop. cDNA synthesis was performed using the RevertAid First Strand cDNA Synthesis Kit (Thermo Scientific). Real-time PCR was performed using a LightCycler^®^96 system (Roche, Basel, Switzerland). SYBR^®^ Green quantitative real-time PCR amplification was performed with Accupower^®^ 2× GreenstarTM qPCR-MasterMix (Bioneer, Deajeon, Korea). The primer sequences were as follows: human TGase 2, forward: 5’-AAT CCC AGG GAA GGA CTT GG-3’ and reverse: 5’-GCA CAT CTA CTG CCT GGT TG-3’; human MDM2, forward: 5’-AGC TCA GCA CTG AGA AGT GT-3’ and reverse: 5’-TCT GAA CAC CTT GCA ACA GC-3’; human GAPDH, forward: 5’-TGA ACG GGA AGC TCA CTG-3’ and reverse: 5’-TCC ACC CTG TTG CTG TA-3’.

### 4.5. VEGF ELISA

Secreted VEGF was measured in the supernatant of cell cultures using the human VEGF Quantikine ELISA Kit (Cat. No. DVE00; R&D Systems, Minneapolis, MN, USA) according to the manufacturer’s instructions. Briefly, 2 × 10^5^ cells were seeded in each well of a 6-well plate and treated at 80% confluence, as described in the figure legends. The supernatant was collected and analyzed. The optical density (O.D.), which indicates VEGF concentration, was measured using an ELISA reader (Thermo Fisher Scientific). The secreted VEGF amount (absorbance) is used to normalize the sulforhodamine B assay (absorbance). Sulforhodamine B (SRB) assay was performed as described [6].

### 4.6. Preclinical Xenograft Tumor Models

Six-week-old, female-specific, pathogen-free BALB/c-nude mice (*n* = 20) were purchased from Central Lab (Animal Inc., Seoul, Korea). Each mouse was injected with CAKI-1 cells (5 × 10^6^ cells/head) subcutaneously. When tumors reached an appropriate size (100–200 mm^3^), the mice were randomized into four groups according to tumor volume and body weight: the control group was treated with vehicle only (0.04% dimethyl sulfoxide [DMSO]) in PBS, the streptonigrin-treated group received 0.1 mg/kg of the compound, and the pazopanib hydrochloride salt-treated group received 100 mg/kg of the compound. Vehicle, streptonigrin, and pazopanib hydrochloride salt were administered orally every other day: Monday, Wednesday, and Friday. The size of the primary tumors was measured every 3–4 days using calipers. Tumor volume was calculated using the following formula: V = (A × B^2^)/2, where V is the volume (mm^3^), A is the long diameter, and B is the short diameter (mm). Mice were euthanized using 7.5% CO_2_, and tumors were harvested for immunohistochemical analysis. This study was reviewed and approved by the Institutional Animal Care and Use Committee (IACUC) of the National Cancer Center Research Institute (NCCRI). The NCCRI is an Association for Assessment and Accreditation of Laboratory Animal Care International (AAALAC International)-accredited facility and abides by the Institute of Laboratory Animal Resources (ILAR) guide (IRB number: NCC-19-464).

### 4.7. Automated Immunohistochemistry

Primary kidney cancer human tissue array (Cat. No. CL) was purchased from SUPER BIO CHIPS (Seoul, Korea). The subtype information is listed in Appendix A. Immunohistochemistry analysis was performed using a VENTANA Discovery XT automated staining instrument (Ventana Medical Systems, Tucson, AZ, USA). Slides were prepared from xenografts and de-paraffinized at 75 °C for 30 min using EZprep solution (Ventana Medical Systems). Epitope retrieval was performed in the automated stainer by exposure to CC1 solution (Ventana Medical Systems) at 95 °C for 64 min. The antibody was titered over a range of concentrations to provide the optimum specific staining to the background staining ratio. Once the titers were set, the antibody was transferred (with diluent) to user-fillable dispensers for use on the automated stainer. Slides were developed using the Optiview DAB detection kit (Ventana Medical Systems). Briefly, the steps were as follows: inhibitor for 8 min, linker for 8 min, multimer for 12 min, DAB/peroxide for 8 min, and copper for 4 min. Then, slides were counterstained with hematoxylin II for 8 min (Ventana Medical Systems). Antibody titers were determined for each antibody using positive and negative control tissues according to the manufacturer’s instructions. Representative images from each tumor were collected using a 20× objective lens.

### 4.8. Statistical Analysis

Statistical analysis of normally distributed data was performed using one-way or two-way analysis of variance followed by Tukey’s multiple comparisons test. All calculations were performed using PRISM 7.0 for Windows.

## 5. Conclusions

This study elucidated the mechanism underlying the effect of TGase 2 on promoting angiogenesis in cancer cells. TGase 2-mediated p53 suppression promoted angiogenesis in RCC by increasing HIF-1α-p300 binding, thereby activating HIF-1α. The TGase 2 inhibitor had a dominant inhibitory effect on anti-angiogenesis over pazopanib by regulating HIF-1α activity. TGase 2 promotes angiogenesis in endothelial cells and in the ECM. A systematic approach based on TGase 2 inhibition may have a synergistic anti-cancer effect by suppressing angiogenesis in several domains, including endothelial cells, the ECM, and cancer cells.

## Figures and Tables

**Figure 1 ijms-21-05042-f001:**
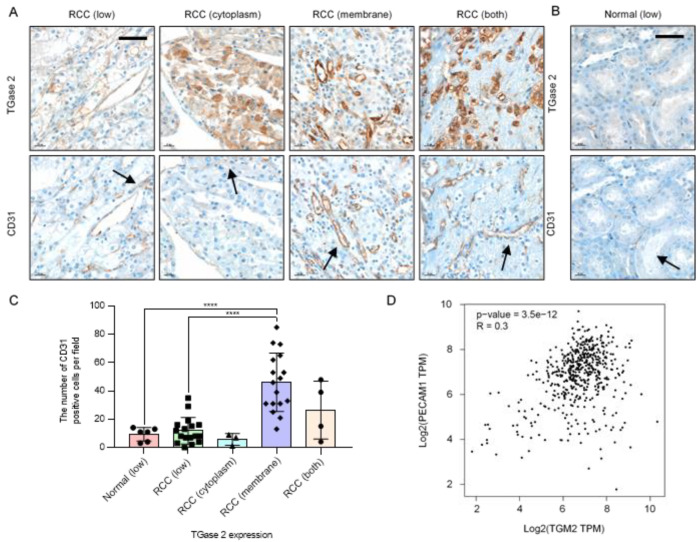
Increased expression of transglutaminase 2 (TGase 2) is associated with increased CD31. (**A**) Representative images of TGase 2 and CD31 (black arrows) staining in a human clear cell renal cell carcinoma (RCC) tissue microarray. (**B**) Representative images of TGase 2 and CD31 (black arrows) staining in human normal kidney tissue. TGase 2 expression was divided into low, high cytoplasm, high cytoplasmic membrane, and both high cytoplasm and cytoplasmic membrane, according to the staining results. (**C**) Average number of CD31-positive cells in the human normal kidney with low TGase 2 expression (*n* = 6), human RCC with low TGase 2 expression (*n* = 17), RCC with high TGase 2 expression in the cytoplasm (*n* = 3), RCC with high TGase 2 expression in the cytoplasmic membrane (*n* = 17), and RCC with high TGase 2 expression in both the cytoplasm and the cytoplasmic membrane (*n* = 4). (**D**) Correlation analysis of genes was conducting using the GEPIA tool. Expressions of *TGM2* (TGase 2 gene) and *PECAM1* (CD31 gene) were positively correlated (*p*-value = 3.5e-12, *R* = 0.3). TPM; transcripts per million reads. Error bars represent SD. GraphPad Prism software was used to perform one-way ANOVA, **** *p* < 0.0001. Scale bar = 50 μm.

**Figure 2 ijms-21-05042-f002:**
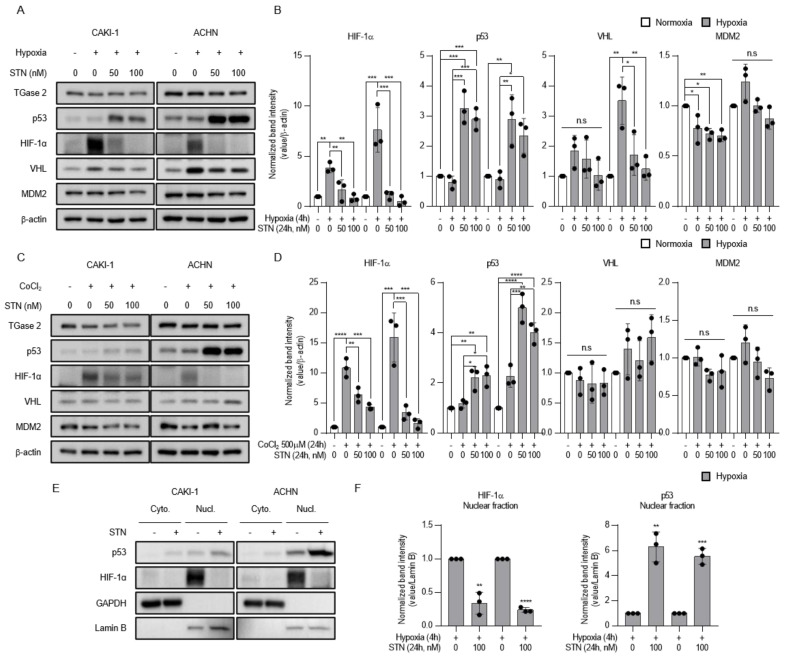
TGase 2 inhibition induces p53-dependent inhibition of hypoxia-inducible factor (HIF)-1α in hypoxia. (**A**) Cells were treated with STN (streptonigrin, TGase 2 inhibitor) for 24 h and incubated for 4h in hypoxia (1% O_2_). (**B**) The image J analysis of Western blotting of Figure 2A. (**C**) Cells were treated with STN and CoCl_2_ (cobalt chloride, 500 μM) and incubated for 24 h in normoxia. Whole cell lysates were subjected to the immunoblotting with indicated antibodies. β-actin was used as a loading control. (**D**) The image J analysis of Western blotting of Figure 2C. (**E**) Cells were treated with or without STN (100 nM) for 24 h and incubated for 4 h in hypoxia (1% O_2_). GAPDH was used as a cytosolic fraction loading control and Lamin B was used as a nuclear fraction loading control. (**F**) The image J analysis of Western blotting of Figure 2E. Densitometry of proteins in nuclear fraction is used to normalize the Lamin B. Error bar represents SD. GraphPad Prism software used to perform one-way ANOVA or t-test, * *p* < 0.05, ** *p* < 0.01, *** *p* < 0.001, **** *p* < 0.0001. ns = not significant. Data are representative of three independent experiments.

**Figure 3 ijms-21-05042-f003:**
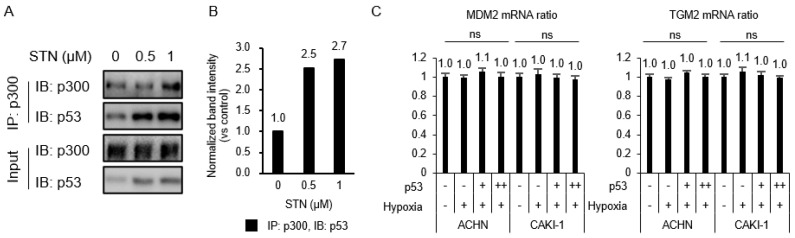
p53–HIF-1α complex is inhibited by TGase 2 inhibition in hypoxia. (**A**) ACHN cells were treated with STN for 2 h and incubated for 4 h in hypoxia (1% O_2_). The proteins were immunoprecipitated from cell extracts using an anti-p300 antibody and subjected to immunoblotting. (**B**) The image J analysis of Western blotting of Figure 3A. (**C**) Cells were transfected with 3xFLAG p53 (p53 overexpression vector; 0, 2, 4 μg) for 8 h and incubated for 4 h in hypoxia. The total RNAs were isolated from cells and RT-PCRs against mouse double minute 2 homolog (MDM2), TGM2, and GAPDH were performed as described in the method. GAPDH was used as an internal control for the RT-PCR system. Error bar represents SD. GraphPad Prism software used to perform one-way ANOVA, ns = not significant. Data are representative of three independent experiments.

**Figure 4 ijms-21-05042-f004:**
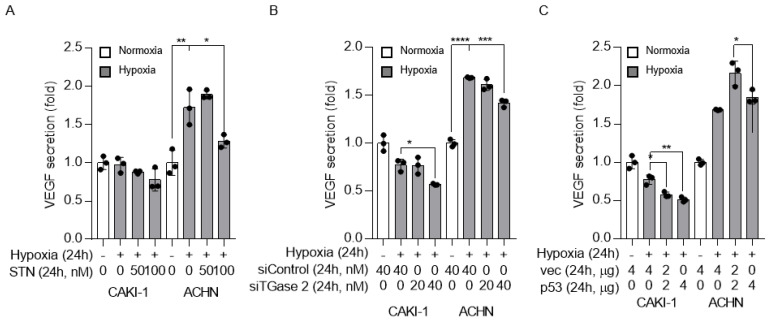
TGase 2 inhibition and p53 overexpression decrease vascular endothelial growth factor (VEGF) secretion under hypoxia. (**A**) VEGF (pg/mL) secretion into the cell culture supernatant was assessed by ELISA. Cells were treated with STN and incubated in serum-free medium under hypoxia (1% O_2_) for 24 h. (**B**) Cells were transfected with control or TGase 2 siRNA and incubated in serum-free medium under hypoxia (1% O_2_) for 24 h. VEGF levels were determined by ELISA. (**C**) Cells were transfected with plasmid encoding wild-type p53 cDNA and incubated in serum-free medium under hypoxia (1% O_2_) for 24 h. VEGF levels were determined by ELISA. The secreted VEGF amount (absorbance) is used to normalize the sulforhodamine B assay (absorbance). Sulforhodamine B assay in cell culture to investigate cell proliferation. Error bar represents SD. GraphPad Prism software used to perform one-way ANOVA, ** *p* < 0.01, *** *p* < 0.001, **** *p* < 0.0001. Data are representative of three independent experiments.

**Figure 5 ijms-21-05042-f005:**
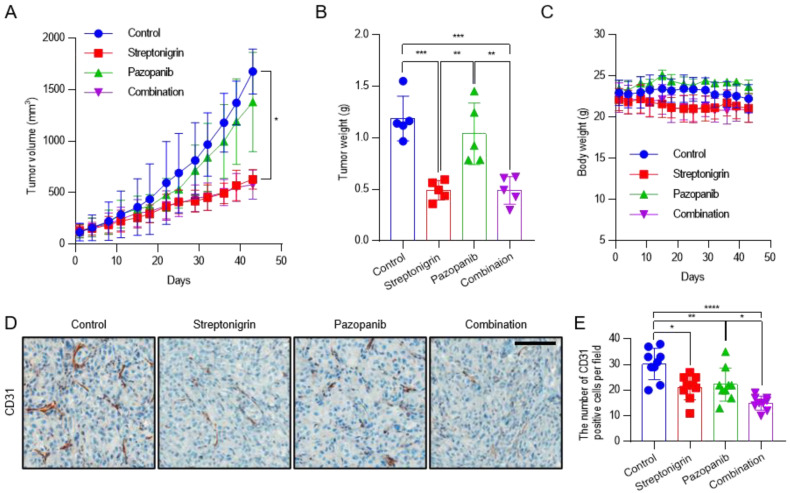
Antitumor activity of pazopanib in combination with TGase 2 inhibition against a CAKI-1 xenograft mouse model. After the subcutaneous injection of CAKI-1 cells (5 × 10^6^ cells), mice were randomized into four groups (*n* = 5) and were treated with vehicle (control), streptonigrin (0.1 mg/kg), or pazopanib (100 mg/kg) as single agents or in combination. Drugs were administered 3 times a week every other day. (**A**) Tumor growth curves during the treatment period. Each point is the mean tumor volume in the group. (**B**) Tumors were isolated from the mice and weighed. (**C**) Body weights of mice during the treatment period. Each point is the mean weight of mice in each group. (**D**) Representative areas of CD31 IHC staining slides in CAKI-1 tumor xenograft from mice. (**E**) Average number of CD31-positive cells in CAKI-1 tumor xenograft from mice (*n* = 9). Error bar represent SD. GraphPad Prism software used to perform one-way or two-way ANOVA, * *p* < 0.05, ** *p* < 0.01, *** *p* < 0.001, **** *p* < 0.0001. Scale bar = 100 μm.

**Figure 6 ijms-21-05042-f006:**
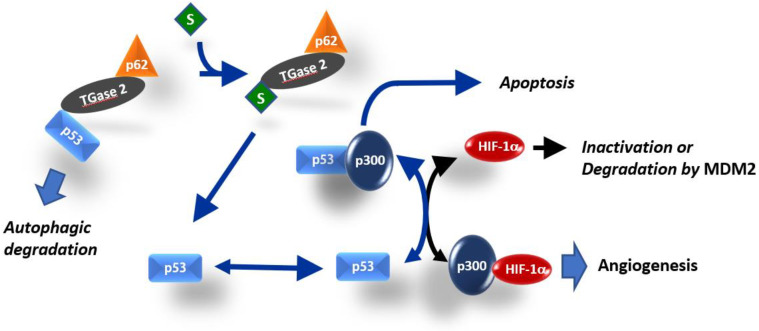
Simplified model for the role of TGase 2 in HIF-1α activation. HIF-1α and p53 both require interaction with the transcriptional co-activator p300 for stable transcriptional activity. Interference of each protein with the transcriptional activity of the other is related to the availability of p300 [18,19]. TGase 2-mediated p53 depletion induces angiogenesis by promoting the HIF-1α–p300 interaction in RCC. The TGase 2 inhibition-mediated increase in apoptosis was attributed to increased p53–p300 binding.

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
