# Peer review of "Transglutaminase 2-Mediated p53 Depletion Promotes Angiogenesis by Increasing HIF-1α-p300 Binding in Renal Cell Carcinoma"

_ijms, 2020, doi:10.3390/ijms21145042_

Round 1

Reviewer 1 Report

Transglutaminase 2-Mediated p53 Depletion Promotes Angiogenesis by Increasing HIF-1α-p300 Binding in Renal Cell Carcinoma by Lee et al.

Strengths:

In this manuscript, the authors performed an extensive experimental analysis to explain the role of Transglutaminase 2 (TGase 2) in the angiogenesis process in renal cell carcinoma (RCC). They demonstrated that TGase 2 inhibition can reverse angiogenesis. This study can be the basis for the development of a new targeted therapy that might have a great impact on RCC patient’s management.

Weaknesses:

  1. Some grammatical errors that can be easily fixed.
  2. Line 108 & 109, (383 cancers reference), this looks confusing, please confirm if you mean 383 references
  3. Line 185 to 195, this paragraph needs to be included in figure 5 legend. Also, I believe (D) should be added instead of (E). In addition (E) needs to be added instead of (F) which should be removed as there is no (F) in the figure.
  4. Line 200 to 201, (The VEGF inhibitor pazopanib did not have a synergistic effect with streptonigrin (Figure 6). Please, explain more why VEGF inhibitor pazopanib did not have a synergistic effect with streptonigrin. Also, I would suggest that figure 6 to be removed from the discussion part and please explain this figure more in the figure legend. Otherwise, this figure might be added as supplementary material.
  5. Line 209, 213 & 222, figure 7 is missing.
  6. Line 226, (reviewed in [8]). Please, remove (reviewed in)

Author Response

Reviewer 1

  1. Some grammatical errors that can be easily fixed.

We have edited by paid experts.

  1. Line 108 & 109, (383 cancers reference), this looks confusing, please confirm if you mean 383 references

We have added the reference No. 6

  1. Line 185 to 195, this paragraph needs to be included in figure 5 legend. Also, I believe (D) should be added instead of (E). In addition (E) needs to be added instead of (F) which should be removed as there is no (F) in the figure.

We have changed the legend of figure 5.

“Figure 5. Antitumor activity of pazopanib in combination with TGase 2 inhibition against CAKI-1 xenograft mouse model. After subcutaneous injection of CAKI-1 cells (5 x 106 cells), mice were randomized in four group (n=5) and were treated with vehicle (control), streptonigrin (0.1 mg/kg), or pazopanib (100 mg/kg) as single agents or in combination. Drugs were administered 3 times a week every other day. (A) Tumor growth curves during the treatment period. Each point is the mean tumor volume in the group. (B) Tumor were isolated from the mice and weighted. (C) Body weights of mice during the treatment period. Each point is the mean weight of mice in each group. (D) Representative areas of CD31 IHC staining slides in CAKI-1 tumor xenograft from mice. (E) Average number of CD31 positive cells in CAKI-1 tumor xenograft from mice (n=9). Error bar represent SD. GraphPad Prism software used to perform one-way or two-way ANOVA, * p<0.05, ** p<0.01, *** p<0.001, ****p<0.0001. Scale bar = 100 μm.”

  1. Line 200 to 201, (The VEGF inhibitor pazopanib did not have a synergistic effect with streptonigrin (Figure 6). 

Please, explain more why VEGF inhibitor pazopanib did not have a synergistic effect with streptonigrinAlso, I would suggest that figure 6 to be removed from the discussion part and please explain this figure more in the figure legend. Otherwise, this figure might be added as supplementary material.

           We have added the result and discussion at result (194-198) as,

“Although the combination treatment was the most effective for reducing tumor angiogenesis, tumor growth by combination treatment of pazopanib and streptonigrin was not reduced synergistically (Figure 5A and E). This suggests that pazopanib decreased tumor angiogenesis without affecting tumor growth, whereas streptonigrin decreased both tumor angiogenesis and tumor growth through p53 activation [6].”

  1. Line 209, 213 & 222, figure 7 is missing.

We have changed the from "Figure 7" to "Figure 6" in the discussion

  1. Line 226, (reviewed in [8]). Please, remove (reviewed in)

We have removed “review in”.

Reviewer 2 Report

The manuscript by Lee et al. “Transglutaminase 2-Mediated p53 Depletion Promotes Angiogenesis by Increasing HIF-1a-p300 Binding in Renal Cell Carcinoma” describes the continuation of Authors studies of the action TGase 2 inhibitor on p53 pathway. The paper provides some level of novelty, the experiments are designed well in general but the study execution/description needs some improvement.

You can find my remarks and suggestions for the Authors below:

Materials and methods:

Please clearly describe the timeline of the in vitro experiments. How long was the exposition to hypoxia/CoCl2/drugs? Where the drugs added simultaneously with pO2 change or you performed some pre-treatment?

The experiments with tissue arrays are not described in the Materials and Methods section. Were all 41 RCC cases from primary tumours? Did the manufacturer provide information on RCC subtype (clear cell, papillary etc) or VHL status? Maybe this could be also correlated with the TGase 2 or CD31 staining?

It is not clear how many repetitions were performed. Did you perform several biological replicates or these are results from one attempt only (as it seems to me) ? Western blot is a semi-quantitative method, so, in my opinion, analyses should be performed on at least 3 independent biological replicates to confirm the results. Especially, as you seem to run loading control (actin) on separate gel from the protein of interest (I concluded that from a different pattern of protein ladder in Supplementary data), which increases the risk of differences of band intensities related to the amount of total protein (do you include Ponceau staining to assure equal loading on each gel at least?). This will allow you to perform statistics on densitometry to allow drawing conclusions about up- or down-regulation of proteins.

No technical information was provided on the overexpression and silencing experiments. Please go through the results again to make sure you provide all information required for the reproducibility of your study.

Results:

All results should be analysed from at least 3 independant biologial replicates with technical replicates and statistical analysis should be clearly marked on figures. 

For the sake of scientific exactness, I think that the densitometry analysis (with statistics) for MDM2, VHL and other proteins that were unchanged by the treatment, should be provided as well (Fig2A-D).

Graph on Fig 2F seems to be a quantification of different results than indicated (of Fig 2E) as it does not show cytoplasmic/ nuclear level of detected proteins. Please make sure you provide correct densitometry results or correct sample description on bars.

Fig 4- did you confirm no effect of the drug/ hypoxia on cell growth to make sure the differences in VEGF secretion are not related to the cell number? In my experience it is safer to normalise the ELISA result for the amount of cells (even to MTT assay if not cell counting). Please provide results of statistical analysis to allow conclusions on increased or decreased secretion.

Discussion:

This section seem a bit too far fetched for me. In line 197 pp7, I would modify "TGase 2 promotes angiogenesis in RCC" to "TGase 2 promotes proangiogenic response in RCC" as the hif/p53 interaction you studied in vitro on cancer cells only and the angiogenesis (endothelial cells) is affected most probably by vegf secretion by these cells. Direct effect of the TGase 2 inhibition on endothelial cells was not checked in this study. 

Over all, you discuss mostly your previously published results. It is difficult to judge what novelty the current study brought. Please relate more to the newly obtained data. Also, which result do you mean when you repeatedly refer Figure 7 (with different references) in the discussion section? 

Author Response

Reviewer 2

Materials and methods:

Please clearly describe the timeline of the in vitro experiments. How long was the exposition to hypoxia/CoCl2/drugs? Where the drugs added simultaneously with pO2 change or you performed some pre-treatment?

We have changed the sentence in the materials and methods

“4.2. Cell Culture

CAKI-1 and ACHN cells were purchased form ATCC (Manassas, VA, USA). Cells were cultured in complete DMEM (Hyclone, Marlborough, MA, USA) containing 10% FBS (Hyclone) and penicillin/streptomycin in an atmosphere of 5% CO2/100% humidity at 37°C. Cells were incubated for 4h under normoxic condition (5% CO2) using conventional incubator or hypoxic conditions (1% O2, 5% CO2, and 94% N2) using Whitley H35 Hypoxystation (Don Whitley Scientific, West Yorkshire, UK). Cells were incubated with or without 500 μM CoCl2 for 24h.”

The experiments with tissue arrays are not described in the Materials and Methods section. Were all 41 RCC cases from primary tumours? Did the manufacturer provide information on RCC subtype (clear cell, papillary etc) or VHL status? Maybe this could be also correlated with the TGase 2 or CD31 staining?

We have change the sentence in the materials, methods and results. RCC subtype was added in the supplementary table 1.

“< Materials and methods >

4.7. Automated Immunohistochemistry

Primary kidney cancer human tissue array (Cat. No. CL) was purchased from SUPER BIO CHIPS (Seoul, KOREA). The subtype information is listed in the supplementary table 1. Immunohistochemistry analysis was performed using a VENTANA Discovery XT automated staining instrument (Ventana Medical Systems, Tucson, AZ, USA). Slides were prepared from xenografts and de-paraffinized at 75°C for 30 min using EZprep solution (Ventana Medical Systems). Epitope retrieval was performed in the automated stainer by exposure to CC1 solution (Ventana Medical Systems) at 95°C for 64 min. The antibody was titered over a range of concentrations to provide the optimum specific staining to background staining ratio. Once the titers were set, the antibody was transferred (with diluent) to user-fillable dispensers for use on the automated stainer. Slides were developed using the Optiview DAB detection kit (Ventana Medical Systems). Briefly, the steps were as follows: inhibitor for 8 min, linker for 8 min, multimer for 12 min, DAB/peroxide for 8 min, and copper for 4 min. Slides were then counterstained with hematoxylin II for 8 min (Ventana Medical Systems). Antibody titers were determined for each antibody using positive and negative control tissues according to the manufacturer’s instructions. Representative images from each tumor were collected using a 20× objective lens.

< Figure 1 and legend is changed>

Figure 1. Increased expression of TGase 2 is associated to increase of CD31.

(A) Representative images with TGase 2 and CD31 (black arrows) staining in human clear cell renal cell carcinoma (RCC) tissue microarray. (B) Representative images with TGase 2 and CD31 (black arrows) staining in human normal kidney tissue. TGase 2 expression was divided into low, high cytoplasm, high cytoplasmic membrane, and both high cytoplasm and cytoplasmic membrane, according to the staining results. (C) Average number of CD31 positive cells in human normal kidney with low TGase 2 expression (n=6), human RCC with low TGase 2 expression (n=17), RCC with high TGase 2 expression in the cytoplasm (n=3), RCC with high TGase 2 expression in the cytoplasmic membrane (n=17) and RCC with high TGase 2 expression in the both cytoplasm and cytoplasmic membrane (n=4). (D) Correlation analysis of genes was conducting using the GEPIA tool. TGM2 and PECAM1 were positively correlated (p-value = 3.5e-12, R = 0.3). TPM; transcripts per million reads. Error bar represent SD. GraphPad Prism software used to perform one-way ANOVA, ****p<0.0001. Scale bar = 50 μm.

< Results >

2.1. TGase 2 expression correlated with expression level of CD31

To test whether TGase 2 is associated with angiogenesis, TGase 2 expression and the number of endothelial cell marker such as CD31 were analyzed. Cases in which TGase 2 was only expressed in the cytoplasmic membrane showed about 4.3 times more CD31 positive cells than normal kidney tissue and RCC with low TGase 2 expression (Figure 1C). As significant correlation was identified in the expression levels of TGase 2 and CD31 in kidney renal clear cell carcinoma, the GEPIA database was used in the present study to analyze the correlations between TGM2 and PECAM1. TGase 2 is encoded by the TGM2 gene and CD31 is encoded by the PECAM1 gene in humans. The results revealed that expression levels of TGM2 and PECAM1 were positively correlated (p-value = 3.5e-12, R = 0.3) (Figure 1D).”

< Supplementary table 1 is added>

Age

Sex

Organ

Diagnosis

TNM

59

Male

Kidney

renal cell carcinoma, clear cell type

T2aN0M0

67

Female

Kidney

renal cell carcinoma, clear cell and granular cell type

T2aNXM0

65

Male

Kidney

renal cell carcinoma, clear cell and granular cell type

T2bNXM0

59

Male

Kidney

renal cell carcinoma, clear cell and granular cell type

T1bNXM0

56

Male

Kidney

renal cell carcinoma, clear cell and pseudosarcomatous type

T1bN0M1

42

Female

Kidney

renal cell carcinoma, clear cell type

T3aNXM0

50

Male

Kidney

renal cell carcinoma, clear cell type

T1aN0M0

60

Female

Kidney

renal cell carcinoma, clear cell type

T3aN0M0

69

Female

Kidney

renal cell carcinoma, clear cell type

T2aNXM0

54

Male

Kidney

renal cell carcinoma, clear cell type

T1bNXM0

43

Female

Kidney

renal cell carcinoma, clear cell type

T1aN0M0

53

Male

Kidney

renal cell carcinoma, clear cell type

T3aN0M1

59

Male

Kidney

renal cell carcinoma, clear cell type

T3aN0M0

61

Male

Kidney

renal cell carcinoma, collecting duct type

T1aNXM0

40

Male

Kidney

renal cell carcinoma, clear cell type

T1aNXM0

64

Male

Kidney

renal cell carcinoma, clear cell type

T1bN0M0

37

Male

Kidney

renal cell carcinoma, collecting duct type

T2bN0M0

65

Male

Kidney

renal cell carcinoma, papillary type

T1bNXM0

74

Male

Kidney

renal cell carcinoma, clear cell type

T3aN0M0

67

Male

Kidney

renal cell carcinoma, clear cell type

T4NXM0

50

Male

Kidney

renal cell carcinoma

T2aN0M0

74

Male

Kidney

renal cell carcinoma, clear cell type

T3aN0M0

62

Male

Kidney

renal cell carcinoma, clear cell and granular cell type

T3aN0M0

45

Male

Kidney

renal cell carcinoma, clear cell type

T1bN0M0

62

Male

Kidney

renal cell carcinoma

T3aN1M0

53

Male

Kidney

renal cell carcinoma, clear cell type

T1bNXM0

64

Male

Kidney

renal cell carcinoma, clear cell type

T1bNXM0

72

Male

Kidney

renal cell carcinoma, clear cell type

T1aNXM0

62

Female

Kidney

renal cell carcinoma, clear cell type

T3aN0M0

56

Male

Kidney

renal cell carcinoma, clear cell type

T3aN0M0

58

Male

Kidney

renal cell carcinoma

T2aN0M0

57

Male

Kidney

renal cell carcinoma, clear cell and granular cell type

T3bN0M0

43

Female

Kidney

renal cell carcinoma, clear cell type

T1bN0M0

52

Female

Kidney

renal cell carcinoma, granular cell type

T1bN0M0

64

Male

Kidney

renal cell carcinoma, granular cell type

T3aN0M0

57

Male

Kidney

renal cell carcinoma, clear cell type

T1bN0M0

59

Male

Kidney

renal cell carcinoma, clear cell and granular cell type

T2aNXM0

47

Male

Kidney

renal cell carcinoma, clear cell type

T1aNXM0

77

Male

Kidney

renal cell carcinoma, clear cell type

T1aN0M0

47

Female

Kidney

renal cell carcinoma, clear cell and granular cell type

T2aNXM0

55

Female

Kidney

renal cell carcinoma, clear cell type

T3aN0M0

Supplementary table 1. Information of human kidney cancer tissue array. Tissues diagnosis were primary kidney renal cell carcinoma, clear cell type or granular cell type. Age was from 37 to 77. Gender was male or female. TNM stage were referred to the AJCC Cancer Staging Manual (7th Edition).

It is not clear how many repetitions were performed. Did you perform several biological replicates or these are results from one attempt only (as it seems to me) ? Western blot is a semi-quantitative method, so, in my opinion, analyses should be performed on at least 3 independent biological replicates to confirm the results. Especially, as you seem to run loading control (actin) on separate gel from the protein of interest (I concluded that from a different pattern of protein ladder in Supplementary data), which increases the risk of differences of band intensities related to the amount of total protein (do you include Ponceau staining to assure equal loading on each gel at least?). This will allow you to perform statistics on densitometry to allow drawing conclusions about up- or down-regulation of proteins.

All data are conducted three independent experiments. Raw data were attached to the supplementary figure.

No technical information was provided on the overexpression and silencing experiments. Please go through the results again to make sure you provide all information required for the reproducibility of your study.

We have changed the sentence in the materials and methods.

4.1. Antibodies and Reagents

Anti-TGase 2 antibody (Cat. No. MA5-12739; 1:1000) was purchased from Sigma-Aldrich (St. Louis, MO, USA). Antibodies against p53 (Cat. No. sc-126; 1:500), MDM2 (Cat. No. sc-813; 1:500), p300 (Cat. No. sc-584; 1:500), Lamin B (Cat. No. sc-6216; 1:1,000), GAPDH (Cat. No. sc-47724; A:1,000), and β-actin (Cat. No. sc-47778; 1:500) were purchased from Santa Cruz Biotechnology (Dallas, TX, USA). Anti-HIF-1α (Cat. No. A700-001; 1:1,000) was purchased from Bethyl laboratories (Montgomery, TX, USA). Anti-VHL (Cat. No. MS-690-P0) was purchased from Thermo Scientific (Waltham, MA, USA). Anti-CD31 (Cat. No. ab28364) was purchased from Abcam (Cambridge, UK). Streptonigrin (Cat. No. S1014) and cobalt chloride (Cat. No. C8661) were purchased from Sigma-Aldrich. Pazopanib hydrochloride salt (Cat. No. P6755) was purchased from LC Laboratories (Woburn, MA, USA). INTERFERin (Cat. No. 409-50) and jetPEI (Cat. No. 101-40N) transfection reagents were from Polyplus (NY, USA). A small interfering RNA (siRNA) duplex targeting human TGase 2 was obtained from GenePharma (Shanghai, CN). TGM2 siRNA, 5’ – UGG GCA GUU UGA AGA UGG GAU CCU A - 3’; and control siRNA, 5’ – CCU CGU GCC GUU CCA UCA GGU AGU U-3’. p3XFLAG p53 plasmid described previously [26].

Results:

All results should be analysed from at least 3 independant biologial replicates with technical replicates and statistical analysis should be clearly marked on figures. 

All data are conducted three independent experiments.

For the sake of scientific exactness, I think that the densitometry analysis (with statistics) for MDM2, VHL and other proteins that were unchanged by the treatment, should be provided as well (Fig2A-D). Graph on Fig 2F seems to be a quantification of different results than indicated (of Fig 2E) as it does not show cytoplasmic/ nuclear level of detected proteins. Please make sure you provide correct densitometry results or correct sample description on bars.

We have changed the figure 2.

Figure 2. TGase 2 inhibition induces p53-dependent inhibition of HIF-1α in hypoxia.

(A) Cells were treated with STN (Streptonigrin, TGase 2 inhibitor) for 24h and incubated for 4h in hypoxia (1% O2). (B) The image J analysis of western blotting of Figure 2A. (C) Cells were treated with STN and CoCl2 (cobalt chloride, 500 μM) and incubated for 24h in normoxia. Whole cell lysates were subjected to the immunoblotting with indicated antibodies. β-actin was used as a loading control. (D) The image J analysis of western blotting of Figure 2C. (E) Cells were treated with or without STN (100 nM) for 24h and incubated for 4h in hypoxia (1% O2). GAPDH was used as a cytosolic fraction loading control and Lamin B was used as a nuclear fraction loading control. (F) The image J analysis of western blotting of Figure 2E. Densitometry of proteins in nuclear fraction is used to normalize the Lamin B. Error bar represents SD. GraphPad Prism software used to perform one-way ANOVA or t-test, *p<0.05, ** p<0.01, *** p<0.001, ****p<0.0001. ns = not significant. Data are representative of three independent experiments.

Fig 4- did you confirm no effect of the drug/ hypoxia on cell growth to make sure the differences in VEGF secretion are not related to the cell number? In my experience it is safer to normalise the ELISA result for the amount of cells (even to MTT assay if not cell counting). Please provide results of statistical analysis to allow conclusions on increased or decreased secretion.

We have changed the figure 4. The secreted VEGF amount (absorbance) is used to normalize the sulforhodamine B assay (absorbance). Sulforhodamine B assay in cell culture to investigate cell proliferation.

Figure 4. TGase 2 inhibition and p53 overexpression decrease VEGF secretion under hypoxia.

(A) VEGF (pg/ml) secretion into the cell culture supernatant was assessed by ELISA. Cells were treated with STN and incubated in serum-free medium under hypoxia (1% O2) for 24 h. (B) Cells were transfected with control or TGase 2 siRNA, and incubated in serum-free medium under hypoxia (1% O2) for 24 h. VEGF levels were determined by ELISA. (C) Cells were transfected with plasmid encoding wild-type p53 cDNA and incubated in serum-free medium under hypoxia (1% O2) for 24 h. VEGF levels were determined by ELISA. The secreted VEGF amount (absorbance) is used to normalize the sulforhodamine B assay (absorbance). Sulforhodamine B assay in cell culture to investigate cell proliferation. Error bar represents SD. GraphPad Prism software used to perform one-way ANOVA, ** p<0.01, *** p<0.001, ****p<0.0001. Data are representative of three independent experiments.

Discussion:

This section seem a bit too far fetched for me. In line 197 pp7, I would modify "TGase 2 promotes angiogenesis in RCC" to "TGase 2 promotes proangiogenic response in RCC" as the hif/p53 interaction you studied in vitro on cancer cells only and the angiogenesis (endothelial cells) is affected most probably by vegf secretion by these cells. Direct effect of the TGase 2 inhibition on endothelial cells was not checked in this study. 

We have changed to “TGase 2 promotes proangiogenic response in RCC”

Over all, you discuss mostly your previously published results. It is difficult to judge what novelty the current study brought. Please relate more to the newly obtained data. Also, which result do you mean when you repeatedly refer Figure 7 (with different references) in the discussion section? 

Figure 7 is Figure 6. We have discussed and summarized the new finding in discussion with Figure 6.
